# StructRAG: Boosting Knowledge Intensive Reasoning of LLMs via Inference-time Hybrid Information Structurization

**Zhuoqun Li**[1,2]**, Xuanang Chen**[1*]**Haiyang Yu**[3]**, Hongyu Lin**[1]**, Yaojie Lu**[1]**, Qiaoyu Tang**[1,2]**,**
**Fei Huang**[3]**, Xianpei Han**[1*]**Le Sun**[1]**, Yongbin Li**[3*]
[1]Chinese Information Processing Laboratory, Institute of Software, Chinese Academy of Sciences
[2]University of Chinese Academy of Sciences
[3]Tongyi Lab
{lizhuoqun2021,chenxuanang,hongyu,luyaojie}@iscas.ac.cn
{tangqiaoyu2020,xianpei,sunle}@iscas.ac.cn
{yifei.yhy,f.huang,shuide.lyb}@alibaba-inc.com

## Abstract

Retrieval-augmented generation (RAG) is a key means to effectively enhance large language models (LLMs) in many knowledge-based tasks. However, existing RAG methods struggle with knowledge-intensive reasoning tasks, because useful information required to these tasks are badly scattered. This characteristic makes it difficult for existing RAG methods to accurately identify key information and perform global reasoning with such noisy augmentation. In this paper, motivated by the cognitive theories that humans convert raw information into various structured knowledge when tackling knowledge-intensive reasoning, we proposes a new framework, StructRAG, which can identify the optimal structure type for the task at hand, reconstruct original documents into this structured format, and infer answers based on the resulting structure. Extensive experiments across various knowledge-intensive tasks show that StructRAG achieves state-of-the-art performance, particularly excelling in challenging scenarios, demonstrating its potential as an effective solution for enhancing LLMs in complex real-world applications.
https://github.com/icip-cas/StructRAG

## 1 Introduction

With the advancement of deep learning technology, large language models (LLMs) have demonstrated considerable strengths in natural language tasks and are extensively applied in complex real-world scenarios (OpenAI et al., 2024; Yang et al., 2024a). However, they still exhibit limitations in factual tasks due to a lack of domain-specific knowledge, real-time updated information, and proprietary knowledge (Huang et al., 2023; Sui et al., 2024). To address this, retrieval-augmented generation (RAG) methods have been developed to effectively provide essential external knowledge (Yu et al., 2022; Gao et al., 2024). Typically, RAG methods involve splitting original documents into shorter chunks, retrieving the most relevant ones based on the query, and then using these chunks to enable LLMs to generate reliable answers (Ma et al., 2023; Li et al., 2024a). Due to their strong performance through this straightforward process, RAG methods are commonly employed in various knowledge-based question-answering tasks (Shi et al., 2024; Wang et al., 2024b).

Unfortunately, current RAG approaches cannot effectively handle knowledge-intensive reasoning tasks due to the scattered nature of related information needed to solve these tasks (Kuratov et al., 2024; Yang et al., 2024b). Specifically, knowledge-intensive reasoning tasks often require a large amount of useful information which is dispersed across many locations in the provided documents, meanwhile the model needs to perform integrated reasoning after retrieving useful information (Yang et al., 2024b; Wang et al., 2024a). Taking *financial report analysis* as an example, given a large number of financial documents and the need to compare the development trends of

---

*Corresponding author.

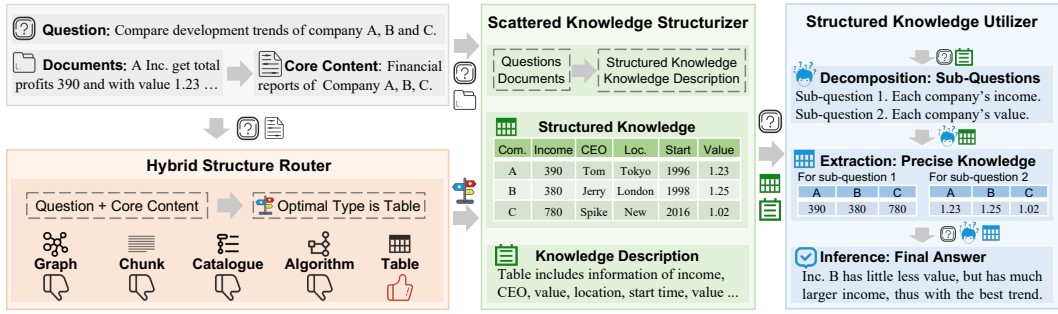

Figure 1: The overview of StructRAG framework, including an hybrid structure router to select the optimal structure type based on task requirements, a scattered knowledge structurizer to convert raw documents into structured knowledge, and a structured knowledge utilizer to decompose complex question and then effectively using the structured knowledge to infer the final answer.

multiple companies, LLMs need to *dig out all relevant financial indicators scattered across original documents and then generate insights by carefully comparing and comprehensively analyzing these indicators*. In such scenarios, standard RAG methods face challenges in accurately retrieving all relevant textual chunks, which may contain substantial noise, and integrating multiple key pieces of information for reasoning, leading to unsatisfactory performance on these tasks.

From a human perspective, people do not solve knowledge-intensive reasoning tasks by simply reading raw texts (Johnson-Laird, 1986; Paivio, 1990). As suggested in cognitive load theory, humans typically summarize scattered information from documents into structured knowledge, which is then used to shorten the reasoning path and enable more accurate judgement (Sweller, 1988; Chandler & Sweller, 1991). Furthermore, cognitive fit theory shows that humans prefer using different types of structured knowledge for various tasks, such as tables for statistical analysis tasks and graphs for long-chain inference (Vessey, 1991; Umanath & Vessey, 1994). In recent years, the rapid development of LLMs has laid the foundation for directly using these models to construct various knowledge structures (Li et al., 2023; Jain et al., 2024). Meanwhile, many studies suggest that LLMs share similarities with humans in how they utilize information and solve complex problems (Wei et al., 2022; Li et al., 2024b). These inspire us to explore whether LLMs can adopt human-like thinking processes to transform scattered information into various structure formats during inference, thereby better serving knowledge-intensive reasoning tasks.

Motivated by this, we propose **StructRAG**, which employs a hybrid information structuring mechanism to construct and utilize structured knowledge in the most suitable format based on task requirements. As illustrated in Figure 1, the StructRAG framework consists of three modules designed to sequentially identify the most suitable structure type, construct structured knowledge in that format, and utilize that structured knowledge to infer the final answer. First, recognizing that different structure types are suited for different tasks, a hybrid structure router is proposed to determine the most appropriate structure type based on the question and document information of the current task. Second, given that constructing structured knowledge is complex and requires strong comprehension and generation abilities, an LLM-based scattered knowledge structurizer is employed to convert raw documents into structured knowledge in the optimal type. Finally, since questions in knowledge-intensive reasoning tasks can often be a complex composite problems that are challenging to solve directly, a structured knowledge utilizer is used to perform question decomposition and precise knowledge extraction for more accurate answer inference.

The core aspect of StructRAG is the hybrid structure router's ability to accurately select the most suitable structure type for each input task. To equip the router with this capability, we propose a training method for the hybrid structure router. Inspired by successful use of reinforcement learning in training LLMs for decision-making tasks (Havrilla et al., 2024; OpenAI, 2024), we employ the DPO algorithm to train the router module, which follows reinforcement learning principles without requiring additional reward models (Rafailov et al., 2023; Allam, 2024). However, there is insufficient training data for the model to learn how to choose the optimal structure type, and collecting enough such data in the real world is also challenging. To address this, we introduce a novel pipeline for constructing preference training data that involves task synthesis, solution simulation, and pref-

erence judgment to create high-quality synthetic data, thereby enhancing the router's ability to select the appropriate structure type.

In our experiments, we evaluate StructRAG across various knowledge-intensive reasoning tasks and compare it with several strong RAG baselines. The results demonstrate that StructRAG achieves state-of-the-art performance, with improvements becoming more pronounced as task complexity increases. This confirms that StructRAG is a robust solution for addressing challenging knowledge-intensive tasks. Additionally, compared to recent Graph RAG methods, StructRAG not only exhibits superior performance across a broader range of tasks but also operates significantly faster on average.

## 2 RELATED WORK

### 2.1 RETRIEVAL-AUGMENTED GENERATION

RAG technology achieves good performance in the era of LLMs by providing external knowledge to assist answering questions and reducing hallucinations (Jiang et al., 2023; Asai et al., 2023; Li et al., 2024c; Gao et al., 2024; Chen et al., 2024). The initial strategy of RAG involves using a retriever to search for and retain highly relevant chunks from a knowledge base based on a query, these chunks are then fed into the generation module as external knowledge, enhancing its performance (Qi et al., 2019; Lewis et al., 2020; Gur et al., 2021; Yu et al., 2022). To improve RAG effectiveness, some approaches have introduced iterative RAG, proposing various enhancements such as query expansion and rewriting (Ma et al., 2023; Li et al., 2024a; Chan et al., 2024; Shi et al., 2024), and others try to improve the corporation between retrieval and generation (Qian et al., 2024; Su et al., 2024; Luo et al., 2024; Zhang et al., 2024). Although existing methods achieve strong performance on multi-hop tasks like HotpotQA, chunk-based RAG struggles with knowledge-intensive tasks (Wang et al., 2024a). This is because chunks must contain excessive text noise and do not capture the interconnections among information , thus LLMs cannot effectively use augmented knowledge.

### 2.2 GRAPH RETRIEVAL-AUGMENTED GENERATION

Recently, to assist LLMs in handling complex question-answering tasks, some works introduce graph structures into RAG systems (Edge et al., 2024; Panda et al., 2024; Peng et al., 2024). One kind of approach uses pre-built knowledge graphs, extracting subgraph based on queries, which are then encoded as soft prompts or flattened into plain text for the generation module (Tang et al., 2024; He et al., 2024; Guan et al., 2024). Another kind of approach involves extracting entity-relation triples from given text documents based on query requirements to construct graph structures, which are then used for knowledge augmentation (Fang et al., 2024; Edge et al., 2024; Panda et al., 2024; Gutierrez et al., 2024). Although these approaches significantly improve performance on multi-hop question-answering tasks, they focus solely on graph-based knowledge via the format of triplets, thus limiting their practical applicability in various domain and application of knowledge-intensive reasoning tasks.

## 3 STRUCTRAG VIA HYBRID INFORMATION STRUCTURIZATION

As mentioned, due to badly dispersed information in knowledge-intensive reasoning tasks, the traditional retrieval module in RAG could retrieve chunks containing substantial textual noise, making it difficult for the generation module to extract useful information for inference. Drawing inspiration from cognitive theories on how humans tackle such tasks, this paper proposes StructRAG, which utilizes a hybrid information structurization mechanism to construct and leverage structured knowledge in its optimal form. Specifically, as illustrated in Figure 1, StructRAG first employs a hybrid structure router to identify the most appropriate structure type for the given task, and then employs a scattered knowledge structurizer to transform raw documents into structured knowledge in that format, and finally incorporates a structured knowledge utilizer to break down complex questions into simpler sub-questions, enabling more accurate reasoning on structured knowledge.

**Task Formulating.** Knowledge-intensive reasoning tasks involved in this paper provide a question $q$ and a large set of documents $D$ as input, with the goal of deriving an answer $a$ based on the

provided documents, which can be expressed as follows:

$$a = \mathcal{F}(q, D), \text{where } D = \{d^{(i)}\}_{i=1}^m \tag{1}$$

where $m$ is the number of documents, which can exceed 20, resulting in a total token of up to 200K. Thus, the most obvious characteristic of these tasks is that useful information is dispersed across the provided documents, requiring the model to engage in complex reasoning based on large-scale relevant data. For example, when comparing the development trends of several companies using a batch of financial reports, the task necessitates retrieving various financial indicators scattered throughout the documents, followed by a detailed comparison of these indicators. This involves considering factors such as the relative importance of different indicators and the magnitude of numerical differences. Consequently, knowledge-intensive reasoning tasks present significant challenges.

**Hybrid Structure Router.** From a human perspective, when solving knowledge-intensive reasoning tasks, individuals tend to use the type of structured knowledge that best matches the specific requirements of faced task. To this end, StructRAG incorporates a hybrid structure router $\mathcal{R}$ to select the optimal structure type. Specifically, the router leverages the question $q$ and the core content $C$ of documents $D$ to make its decision and generate the most suitable structure type $t$, as it is impractical to process the entire set of documents at once.

$$t = \mathcal{R}(q, C), \text{where } C = \{c^{(i)}\}_{i=1}^m \tag{2}$$

The core content $C$ is the concentrate of the titles or the first few sentences from each document $d^{(i)}$. In our work, there are five candidate structure types for five kinds of knowledge-intensive tasks: table for statistical tasks, graph for long-chain tasks, algorithm for planning tasks, catalogue for summarizing tasks, and chunk for simple single-hop tasks. Considering the core effect of the router in the overall framework, our work designs a DPO-based training method to develop a router that excels in knowledge type decision, which is detailed in Section 4.

**Scattered Knowledge Structurizer.** After identifying the most suitable structure type, StructRAG extracts the textual knowledge scattered across raw documents and reconstructs it into structured knowledge. This process requires a comprehensive understanding of all raw documents and and precise formatting of the information, making it a challenging and flexible problem. Therefore, StructRAG employs an LLM-based scattered knowledge structurizer to facilitate the structurization process. Specifically, as shown in Eq. 3 the structurizer $\mathcal{S}$ takes the question $q$, the selected type $t$, and each raw document $d^{(i)}$ as input, and extract the structured knowledge $k_t^{(i)}$ from the document via the powerful understanding and generation ability of LLMs. In addition, a description of the structured knowledge $k_t$ is also generated.

$$k_t^{(i)}, b_t^{(i)} = \mathcal{S}(q, t, d^{(i)}) \tag{3}$$

After that, all output structured knowledge will be collected as the overall one $K_t = \{k_t^{(i)}\}_{i=1}^m$, and the overall description of the whole structured knowledge is constructed as $B_t = \{b_t^{(i)}\}_{i=1}^m$. In term of detailed representation of each kind of structure, the table is by markdown, graph by in list of head-relationship-tail triplets, chunk is by regular text, algorithm is by pseudo code, and catalogue is by text with hierarchical number (e.g., *Section One, 1.1, 1.1.2*) as explicitly chapter identifier.

**Structured Knowledge Utilizer.** After obtaining the structured knowledge in its optimal type, StructRAG performs reasoning to answer the question. Given that the question can be highly combinatorial, this may hinder the identification and use of relevant information in the structured knowledge. Therefore, StructRAG employs an LLM-based structured knowledge utilizer to facilitate question decomposition, precise knowledge extraction, and final answer inference. Specifically, the decomposition process of the utilizer takes the original question $q$ and the overall description of structured knowledge $B_t$ as input, breaking the question down into several simple and intuitive sub-questions $\hat{q}^{(j)}$. Next, the extraction process aims to find out precise knowledge $\hat{k}_t^{(j)}$ for each sub-question $\hat{q}^{(j)}$ from the whole structured knowledge $K_t$. Finally, the inference process integrates all the sub-questions and their extracted precise knowledge to generate a final answer $a$, which can be expressed as follows:

$$\hat{Q} = \mathcal{U}_{\text{decompose}}(q, B_t) = \{\hat{q}^{(j)}\}_{j=1}^n \tag{4}$$

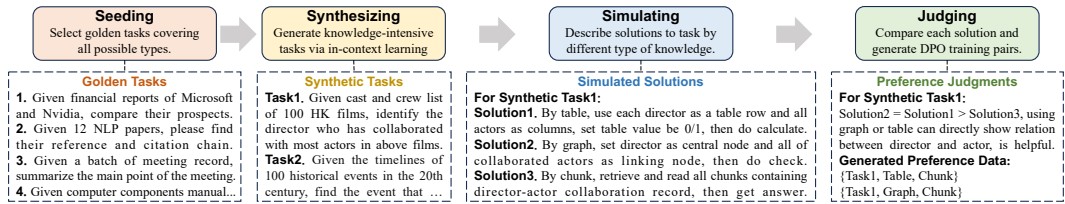

Figure 2: The illustration of training data constructing. First use LLMs to synthesize knowledge-intensive tasks, and then simulate solutions by structured knowledge in different types, finally judge all possible solutions and get preference pairs about candidate structure types.

$$\hat{K}_t = \{\mathcal{U}_{\text{extract}}(\hat{q}^{(j)}, K_t)\}_{j=1}^n = \{\hat{k}_t^{(j)}\}_{j=1}^n \tag{5}$$

$$a = \mathcal{U}_{\text{infer}}(q, \hat{Q}, \hat{K}_t) \tag{6}$$

where $n$ is number of sub-questions, $\hat{Q}$ is set of all sub-questions, $\hat{K}_t$ is whole precise knowledge for all sub-questions, and $\mathcal{U}_{\text{decompose}}$, $\mathcal{U}_{\text{extract}}$ and $\mathcal{U}_{\text{infer}}$ are process of decomposition, extraction and inference, respectively. More details about the utilizer are shown in Appendix A.3.

## 4 Hybrid Structure Router Training

In the StructRAG framework described above, the core factor is accurately determining the most suitable structure type based on the input task, and the performance of the hybrid structure router directly influences the overall effectiveness of the framework. Therefore, to achieve a high-performance router, we propose a training method to enhance the ability of LLMs in identifying the suitable structure type of knowledge. Specifically, given the strong capabilities of reinforcement learning in decision-making scenarios, we train the router using the DPO algorithm, which achieves results similar to reinforcement learning while avoiding the need for additional reward models. Regarding training data, since there is no existing preference data for the optimal structure type selection task, we design a synthesizing-simulating-judging method to efficiently construct preference pairs for training. A detailed explanation is provided in the following paragraphs, along with examples and prompts in the Appendix A.1.

**Data Constructing.** Due to the scarcity of training data for selecting the optimal structure type in the current NLP community, we employ a synthesizing-simulating-judging method to construct preference pairs for training the router. Specifically, as illustrated in Figure 2, given several manually collected seed tasks that covering the possible structure types, we first use LLMs to synthesize a set of new tasks by the in-context learning method, where each task contains a question and core context for documents. Then, for each synthetic task, LLMs is employed to simulate the process of addressing this task by structured knowledge in different types, thus getting different simulated solutions. Finally, a LLM-based judge compares these simulated solutions for solving the task, generating preference pairs regarding the structure types. Each constructed data entry includes a question, the core contents of documents, the chosen structure type, and the rejected structure type, as expressed as follows:

$$D_{\text{synthetic}} = \{q^{(k)}, C^{(k)}, t_w^{(k)}, t_l^{(k)}\}_{k=1}^N \tag{7}$$

where $t_w$ and $t_l$ are chosen structure type and rejected structure type, respectively. In addition, synthetic preference pairs include both English and Chinese data in order to improve the universality.

**Preference Training.** Inspired by the success of using reinforcement learning to train LLMs for decision-making tasks, we employ the DPO algorithm to train the router module, which can get the same effectiveness as reinforcement learning without adding additional reward models. Specifically, the input for training the router includes a question and the core contents of documents, and the output is one kind of structure type (e.g., "table", "graph"). As described in last paragraph, we simulate and construct a set of preference pairs for DPO training, which can be formulated as following:

$$\mathcal{L}_{\text{DPO}}(\pi_\theta; \pi_{\text{ref}}) = -\mathbb{E}_{(q, C, t_w, t_l) \sim D_{\text{synthetic}}} \left[ \log \sigma \left( \beta \log \frac{\pi_\theta(t_w \mid q, C)}{\pi_{\text{ref}}(t_w \mid q, C)} - \beta \log \frac{\pi_\theta(t_l \mid q, C)}{\pi_{\text{ref}}(t_l \mid q, C)} \right) \right] \tag{8}$$

where $\pi_\theta$ and $\pi_{\text{ref}}$ are target policy and reference policy, respectively, and $\beta$ is a hyperparameter. As analyzed later, this preference training enables the model to differentiate between various types of knowledge and their suitability for a given task, resulting in better performance compared to zero-shot and few-shot settings.

# 5 EXPERIMENTS

## 5.1 EXPERIMENTAL SETTINGS

**Evaluation Datasets.** This paper includes various knowledge-intensive reasoning tasks in evaluation. First, this paper chooses the **Loong** benchmark (Wang et al., 2024a), which includes four tasks (Spotlight Locating, Comparison, Clustering, and Chain of Reasoning) and four document length settings, as the length of the document increases, the useful information needed to solve the task becomes more dispersed. As for metrics, this paper adheres to original settings in Loong and use the official code repository[1] in evaluation, involving using LLMs to decide a score from 0 to 100 and the exact matching (EM) rate. In addition, this paper chooses **Podcast Transcripts**, which is a query-focused summarization task reported by GraphRAG (Edge et al., 2024). As for metrics, this paper follows GraphRAG settings, involving head-to-head win rate by a LLM judgement, across four kinds of dimension, which are comprehensiveness, diversity, empowerment, and directness.

**Implementation Details.** We build framework based on Qwen2 series models (Yang et al., 2024a). For the hybrid structure router, StructRAG uses Qwen2-7B-Instruct as the base model and implement DPO training by trl[2]. As for the details of hybrid structure router training, StructRAG constructs and uses a total of 900 preference data, setting the learning rate as 1e-5, number of epochs as 3 and the $\beta$ as default in training. For the scattered knowledge structurizer and strutured knowledge utilizer, StructRAG directly uses Qwen2-72B-Instruct as base model and deploy models as API using vllm[3] following the same setting as in Loong (Wang et al., 2024a).

**Selected Baselines.** We select baselines from commonly used or recent methods for knowledge-based question-answering tasks. Specifically, (1) **Long-Context** (Yang et al., 2024a), which extends the input window of LLMs to up to 128K tokens through extrapolation techniques, allowing large-scale documents to be directly input into the model. (2) **RAG** (Lewis et al., 2020), which splits the given documents into multiple short chunks and uses a retriever to retain only the most relevant chunks as augmentation based on the question. (3) **RQ-RAG** (Chan et al., 2024), which uses a trained LLM to decompose and refine the original question to more accurately find the required chunk augmentation. (4) **GraphRAG** (Edge et al., 2024), which extracts triples (head, relationship, tail) from raw documents and constructs into multi-layered graphs, then uses structured information in graphs to help the generation model answer questions. In implement, for Long-Context and RAG, we directly follow the experimental settings reported in Loong (Wang et al., 2024a), and for RQ-RAG[4] and GraphRAG[5], we evaluate the performance based on the official code repositories. Noting that, for fair comparison, we also set Qwen-72B-Instruct as base model of GraphRAG.

## 5.2 OVERALL RESULTS

Results compared with baselines are shown in Table 1 and Table 2, there are two main conclusions:

**1) StructRAG is a powerful solution to addressing knowledge-intensive reasoning tasks.** Based on the experimental results in Table 1, StructRAG outperforms the baselines in most tasks and document length settings, and in the overall metric, StructRAG exceeds all baselines in both LLM score and EM rate. In addition, as shown in Table 2, StructRAG achieves the best average performance compared to all baselines in Podcast Transcripts. All in all, these experimental findings demonstrate that StructRAG can effectively address knowledge-intensive reasoning tasks and improve a lot compared with previous long-context methods, and different kind of existing powerful RAG techniques.

---

[1]https://github.com/MozerWang/Loong

[2]https://github.com/huggingface/trl

[3]https://pypi.org/project/vllm/

[4]https://github.com/chanchimin/RQ-RAG

[5]https://pypi.org/project/graphrag/

| Method | Spot. | | Comp. | | Clus. | | Chain. | | Overall | |
|---|---|---|---|---|---|---|---|---|---|---|
| | LLM Score | EM | LLM Score | EM | LLM Score | EM | LLM Score | EM | LLM Score | EM |
| Set 1 (10K-50K Tokens) | | | | | | | | | | |
| Long-context (Yang et al., 2024a) | 68.49 | **0.55** | 60.60 | 0.37 | 47.08 | 0.08 | **70.39** | **0.36** | 60.11 | 0.29 |
| RAG (Lewis et al., 2020) | 51.08 | 0.35 | 44.53 | 0.27 | 37.96 | 0.05 | 53.95 | 0.35 | 46.11 | 0.23 |
| RQ-RAG (Chan et al., 2024) | 72.31 | 0.54 | 48.16 | 0.05 | 47.44 | 0.07 | 58.96 | 0.25 | 53.51 | 0.17 |
| GraphRAG (Edge et al., 2024) | 31.67 | 0.00 | 27.60 | 0.00 | 40.71 | 0.14 | 54.29 | 0.43 | 40.82 | 0.18 |
| StructRAG (Ours) | **74.53** | 0.47 | **75.58** | **0.47** | **65.13** | **0.23** | 67.84 | 0.34 | **69.43** | **0.35** |
| Set 2 (50K-100K Tokens) | | | | | | | | | | |
| Long-context (Yang et al., 2024a) | 64.53 | 0.43 | 42.60 | 0.21 | 38.52 | 0.05 | 51.18 | 0.20 | 45.71 | 0.17 |
| RAG (Lewis et al., 2020) | 66.27 | **0.46** | 46.28 | 0.31 | 38.95 | 0.05 | 46.15 | **0.22** | 45.42 | 0.19 |
| RQ-RAG (Chan et al., 2024) | 57.35 | 0.35 | 50.83 | 0.16 | 42.85 | 0.03 | 47.60 | 0.10 | 47.09 | 0.10 |
| GraphRAG (Edge et al., 2024) | 24.80 | 0.00 | 14.29 | 0.00 | 37.86 | 0.00 | 46.25 | 0.12 | 33.06 | 0.03 |
| StructRAG (Ours) | **68.00** | 0.41 | **63.71** | **0.36** | **61.40** | **0.17** | **54.70** | 0.19 | **60.95** | **0.24** |
| Set 3 (100K-200K Tokens) | | | | | | | | | | |
| Long-context (Yang et al., 2024a) | 46.99 | 0.27 | 37.06 | 0.13 | 31.50 | 0.02 | 35.01 | 0.07 | 35.94 | 0.09 |
| RAG (Lewis et al., 2020) | **73.69** | **0.55** | 42.20 | 0.27 | 32.78 | 0.02 | 37.65 | 0.13 | 42.60 | 0.18 |
| RQ-RAG (Chan et al., 2024) | 50.50 | 0.13 | 44.62 | 0.00 | 36.98 | 0.00 | 36.79 | 0.07 | 40.93 | 0.05 |
| GraphRAG (Edge et al., 2024) | 15.83 | 0.00 | 27.40 | 0.00 | 42.50 | 0.00 | 43.33 | 0.17 | 33.28 | 0.04 |
| StructRAG (Ours) | 68.62 | 0.44 | **57.74** | **0.35** | **58.27** | **0.10** | **49.73** | **0.13** | **57.92** | **0.21** |
| Set 4 (200K-250K Tokens) | | | | | | | | | | |
| Long-context (Yang et al., 2024a) | 33.18 | 0.16 | 26.59 | 0.08 | 29.84 | **0.01** | 25.81 | 0.04 | 28.92 | 0.06 |
| RAG (Lewis et al., 2020) | 52.17 | **0.24** | 24.60 | 0.10 | 26.78 | 0.00 | 17.79 | 0.00 | 29.29 | 0.07 |
| RQ-RAG (Chan et al., 2024) | 29.17 | 0.08 | 40.36 | 0.00 | 26.92 | 0.00 | 34.69 | 0.00 | 31.91 | 0.01 |
| GraphRAG (Edge et al., 2024) | 17.50 | 0.00 | 26.67 | 0.00 | 20.91 | 0.00 | 33.67 | 0.33 | 23.47 | 0.05 |
| StructRAG (Ours) | **56.87** | 0.19 | **55.62** | **0.25** | **56.59** | 0.00 | **35.71** | 0.05 | **51.42** | **0.10** |

Table 1: LLM-judged scores and exact matching rate in knowledge-intensive tasks of Loong benchmark. From Set 1 to Set 4, task complexity gradually increases, as reflected by the growing token number of documents. The table show two main conclusions: StructRAG get the SOTA performance in overall metrics. And the more complex the task, the greater the improvement of StructRAG.

| Compared Method Pair | Comprehensiveness | Diversity | Empowerment | Directness | Average |
|---|---|---|---|---|---|
| StructRAG vs. Long-context (Yang et al., 2024a) | 98 | 96 | 97 | 92 | 95.75 |
| StructRAG vs. RAG (Lewis et al., 2020) | 67 | 78 | 51 | 52 | 62.00 |
| StructRAG vs. RQ-RAG (Chan et al., 2024) | 67 | 75 | 50 | 46 | 59.50 |
| StructRAG vs. GraphRAG (Edge et al., 2024) | 61 | 68 | 42 | 41 | 53.00 |

Table 2: Win rate of StructRAG vs. baselines on Podcast Transcripts. StructRAG achieves the best average performance compared to all baselines, further conforming effectiveness of framework.

**2) StructRAG is particularly suitable for complex tasks, performance improvement becomes more significant in scenarios with more dispersed information.** Based on the overall performance in Table 1, the performance comparison between StructRAG and the long-context baseline shows that StructRAG achieves performance improvements of approximately 9, 15, 22, and 23 on Set 1, Set 2, Set 3, and Set 4, respectively. Similarly, comparing StructRAG with RAG shows performance improvements of around 15, 15, and 22 on Set 2, Set 3, and Set 4, respectively. Each set represents the total length of the given documents, with Set 1 being the shortest and Set 4 being the longest. This means that the information needed to answer the questions becomes more dispersed as the length of the documents increases, and making the reasoning process more challenging. Therefore, these results indicate that StructRAG shows more significant improvements over the baselines with longer documents and more scattered information, demonstrating that abilities of our framework to construct and use the optimal type of structured knowledge is especially effective for complex tasks.

## 5.3 ABLATION RESULTS OF MODULES

To validate the role of each module in StructRAG, we perform ablation experiments. As shown in Table 3, "w/o router" refers to random routing, "w/o structurizer" means using only chunks, and "w/o utilizer" refers to directly concatenating the structured knowledge with the original question for answer generation. There are following conclusions:

**1) All three modules contribute positively to the overall framework.** The table shows that removing any of the three modules results in a noticeable performance decline. The overall performance will reduce from 60.38 to 45.33, 53.92 and 55.94 for the router, structurizer, and utilizer, respec-

| Method | Set 1 | | Set 2 | | Set 3 | | Set 4 | | Overall | |
|---|---|---|---|---|---|---|---|---|---|---|
| | LLM Score | EM | LLM Score | EM | LLM Score | EM | LLM Score | EM | LLM Score | EM |
| StructRAG | **69.43** | **0.35** | **60.95** | **0.24** | **57.92** | **0.21** | **51.42** | **0.10** | **60.38** | **0.23** |
| w/o router | 51.09 | 0.28 | 48.28 | 0.17 | 39.52 | 0.13 | 41.83 | **0.10** | 45.33 | 0.17 |
| w/o structurizer | 64.97 | 0.29 | 52.17 | 0.17 | 53.18 | 0.19 | 44.24 | **0.10** | 53.92 | 0.19 |
| w/o utilizer | 68.23 | 0.29 | 59.73 | **0.24** | 53.29 | 0.19 | 35.77 | **0.10** | 55.94 | 0.22 |

Table 3: Ablation results of three modules. The table shows that all three module are with positive contribution, and the most core module is the hybrid structure router.

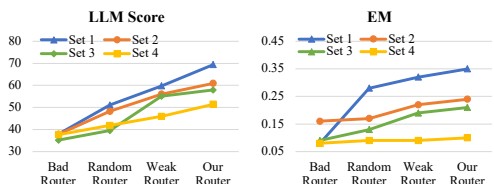

Figure 3: Performance of StructRAG with different routers. The strong router shows obvious positive impact on the overall framework.

| Method | F1 | ACC |
|---|---|---|
| Hybrid Structure Router | **93.42** | **94.38** |
| Qwen2-7B-Instruct (zero-shot) | 50.04 | 54.25 |
| Qwen2-7B-Instruct (few-shot) | 65.59 | 66.12 |
| Qwen2-72B-Instruct (zero-shot) | 78.38 | 80.56 |
| Qwen2-72B-Instruct (few-shot) | 89.39 | 90.06 |

Table 4: Results of evaluating hybrid structure routers. The table shows that preference training is necessary for the routing ability.

tively. This proves that all three modules play an irreplaceable role, and StructRAG tightly and orderly combines these three modules to achieve excellent overall performance.

**2) Choosing the suitable structure type and constructing documents into structured knowledge are more crucial than designing complex utilization methods.** A comparison in the table reveals that different modules are with different importance. The performance drop is most significant when the router is removed, with a decreasing from 60.38 to 45.33. In contrast, removing the utilizer leads to a smaller performance decline, from 60.38 to 55.94. This indicates that simply question-refining as existing methods provides limited improvement for knowledge-intensive reasoning tasks, a more promising direction is constructing and using structured knowledge in suitable type.

## 5.4 DETAILED ANALYSIS

In this section, we do some detailed analysis, including, performance and impact of the router, drawback of using fixed structure type, case study about EM rate performance, and efficiency reports.

### 5.4.1 EFFECT OF THE ROUTER

To explore the necessity of constructing data and conducting DPO training, and relationship between performance of hybrid structure router and overall StructRAG, we first compare our router with raw LLMs, and then draw curl of router and overall StructRAG score. There are following conclusions:

**1) Selecting the optimal type of knowledge based on the task is challenging for raw LLMs without special training.** Based on the experimental results in Table 4, the router trained based on Qwen2-7B-Instruct model significantly outperforms the 72B model with few-shot setting. This indicates that LLMs need some special training to get the ability of selecting the optimal structure type based on needs of the task, even when the model scale reaches 72B.

**2) The performance of hybrid structure router is with significant relevance with the final performance of StructRAG.** As shown in Figure 3, we select Qwen2-72B-Instruct (zero-shot) as the weak router, and design a completely random router and a completely incorrect bad router. The curves in the figure clearly show a positive correlation between router accuracy and the overall performance of the StructRAG framework. This further demonstrates that selecting knowledge types that match the task needs for augmentation is crucial in knowledge-intensice reasoning tasks.

| Method | Set 1 | | Set 2 | | Set 3 | | Set 4 | | Overall | |
|---|---|---|---|---|---|---|---|---|---|---|
| | LLM Score | EM | LLM Score | EM | LLM Score | EM | LLM Score | EM | LLM Score | EM |
| StructRAG | **69.43** | **0.35** | **60.95** | **0.24** | **57.92** | **0.21** | **51.42** | 0.10 | **60.38** | **0.23** |
| w/ only table | 48.00 | 0.23 | 55.19 | **0.24** | 50.35 | 0.19 | 38.44 | **0.12** | 49.66 | 0.21 |
| w/ only graph | 30.59 | 0.09 | 24.05 | 0.05 | 17.46 | 0.03 | 20.96 | 0.04 | 22.71 | 0.05 |
| w/ only chunk | 64.97 | 0.29 | 52.17 | 0.17 | 53.18 | 0.19 | 44.24 | 0.10 | 53.92 | 0.19 |
| w/ only catalogue | 30.49 | 0.10 | 36.36 | 0.13 | 36.77 | 0.12 | 23.75 | 0.03 | 33.26 | 0.10 |
| w/ only algorithm | 43.53 | 0.24 | 32.86 | 0.08 | 31.59 | 0.13 | 16.67 | 0.04 | 32.32 | 0.12 |

Table 5: Results of only containing structured knowledge in one fixed type. It shows any single fixed type is insufficient, confirming the advance of StructRAG via hybrid information structurization.

| Method | Constructing | Reading | Total Latency |
|---|---|---|---|
| RQ-RAG | 7.8 | 1.2 | 9.0 |
| GraphRAG | 215.3 | 1.8 | 217.1 |
| StructRAG (Ours) | 8.2 | 1.5 | 9.7 |

Figure 4: Comparison of implementing latency (minute). The StructRAG has an available speed, which is a little slower than RQ-RAG, but is much faster than GraphRAG method.

| Raw | depreciation of **$ 1,308,463** and share-based compensation expense of **$ 537,197**) in 2024 ... |
|---|---|
| **Structured** | Year. / Com. / CEO / Depreciation / Compensation
2024 / A / Judy / 1308463 / 537197 |

Figure 5: Some cases to show the reason why EM rate is not perfect, because of a little difference in textual format after structurization.

### 5.4.2 DRAWBACK OF A FIXED TYPE OF KNOWLEDGE

To further verify the importance of containing hybrid types of structure rather than a fixed type, we freeze the structure type used in the framework as either chunk, graph, table, algorithm or catalogue for all evaluation tasks. There are the following conclusions:

**Using a single fixed type of knowledge cannot achieve good performance on diverse tasks.** Based on the experimental results in Table 5, it shows that for both scores and exact matching rate, using a single fixed type performs worse than selecting the optimal structure type for needs of the input task. In comparison, the performance degradation is least when only using chunk, with a reduction from 60.38 to 53.92, in other cases, the performance decline is more significant. This can demonstrate the effectiveness of cognitive fit theory in LLMs, meaning that using structured knowledge in the most suitable type can effectively enhance problem-solving abilities.

### 5.4.3 CASE STUDY ABOUT EM METRIC

According to the experimental results in Table 1, StructRAG surpasses baselines in general score, but falls short in seven sub-situations for the exact matching rate. Therefore, we analysis some cases that StructRAG method gets high score but fails exact matching. The reason is mainly about structurization process may alter the textual format of original information. As shown in Table 5, there are some wording differences between structured knowledge and raw information (e.g. from original *"$ 1,308,463"* to *"138463"* in the table). Intuitively this aligns with the common sense, where structurizer is a probabilistic language model rather than rule-based model, thus some possible textual loss may be unavoidable, and output from GraphRAG method also show similar issue.

### 5.4.4 EFFICIENCY REPORT

In this section, we report average latency of StructRAG and compare it with the RQ-RAG (Chan et al., 2024) and GraphRAG (Edge et al., 2024). The latency includes two components: The first part is constructing latency, referring to the process of iteratively retrieving chunks for RQ-RAG, constructing graphs for GraphRAG, and determining the optimal knowledge type and constructing corresponding structure for StructRAG. The second part is reading latency, referring to the process of using augmented knowledge to generate final answers. As shown in Table 4. StructRAG has slightly higher latency compared to RQ-RAG but is obviously faster than GraphRAG. Therefore, StructRAG is a kind of high-performance framework with available implementing speed.

## 6 CONCLUSION

In this paper, noticed the limitation of existing RAG methods in knowledge-intensive reasoning tasks, and inspired by cognitive theories about how human beings solve such tasks, we propose a new framework, StructRAG via hybrid information structurization, which can construct and utilize structured knowledge in the optimal format as augmentation. StructRAG includes a hybrid structure router to precisely select the optimal structure type, then a scattered knowledge structurizer to convert raw documents into structured knowledge, and finally a structured knowledge utilizer to decompose complex questions and infer the final answer via the constructed structured knowledge. Furthermore, in order to get a high-performance hybrid structure router, we construct training data by a synthesizing-simulating-judging pipeline and then implement preference training via DPO algorithm. Experiments on extensive knowledge-intensive reasoning tasks demonstrate that StructRAG is an effective solution, which reach the SOTA performance and can achieve large improvement in badly information-scattered scenarios. Therefore, this paper offers a promising direction, focused on hybrid structured knowledge, for developing more powerful RAG systems in the future.

## ACKNOWLEDGMENTS

We sincerely thank the reviewers for their insightful comments and valuable suggestions. This work was supported by the Natural Science Foundation of China (No. 62122077), Beijing Natural Science Foundation (L243006), Beijing Municipal Science and Technology Project (Nos. Z231100010323002). This work was supported by Alibaba Innovative Research Program.

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

Examples:
########
Knowledge Info:
We have chunks of "Judgment Document 7" \n "Judgment Document 3" \n "Judgment Document 2" \n "Judgment Document 4" \n "Judgment Document 6" \n "Judgment Document 8" \n "Judgment……

Query:
Please classify the above judgment documents according to the six categories: 'Ownership Dispute', 'Administrative Body - Labor and Social Security Administration (Labor, Social Security)', 'Execution Cause - Non-litigation Administrative Execution', 'Embezzlement and Bribery', 'Execution Cause - Other Causes', and 'Administrative Action - Administrative Payment'. Just output the title of each judgment document. Please respond in the format provided, and titles should correspond to the actual judgment documents: \n{'Ownership Dispute': ['"Judgment Document a"', '"Judgment Document b"'], 'Administrative Body - Labor and Social Security Administration (Labor, Social Security)': ['"Judgment Document a"', '"Judgment Document b"'], …….

Output:
Determine whether the given judgment document's cause is 'Ownership Dispute', determine whether the given judgment document's cause is 'Administrative Body - Labor and Social Security Administration (Labor, Social Security)', determine whether the given judgment document's cause is 'Execution Cause - Non-litigation Administrative Execution', determine whether the given judgment document's cause is 'Embezzlement and Bribery', determine whether the given judgment document's cause is 'Execution Cause - Other Causes', determine whether the given judgment document's cause is 'Administrative Action - Administrative Payment'.
########

Knowledge Info: {knowledge_info}

Query: {query}

Output:

---

**Prompts in Extracting Precise Knowledge**

Instruction:
Extract precise from structured knowledge based on the question.

Sub-question: {sub_question}
Structured Knowledge: {knowledge}

Output:

---

**Prompts in Inferencing Final Answer**

Instruction:
Merging all sub-questions and precise knowledge to get final answer.

Question: {question}
Sub-questions {sub_questions}
Precise Knowledge: {precise_knowledge}

Output:

---

Figure 6: Prompts used in structured knowledge utilizer, including decomposing sub-questions, extracting precise knowledge, and infering final answer.

## A    PROMPTS IN STRUCTRAG

### A.1    PROMPTS IN DATA CONSTRUCTING

Considering the current lack of training data that determines the optimal structure type based on task requirements, and the difficulty of collecting such data in real-world scenarios, we designed a pipeline for synthesizing, simulating, and judging data for DPO training specifically aimed at a hybrid structure router. In terms of implementation, we created several prompts to drive the three processes of the LLM, as shown in Figure 8.

### A.2    PROMPTS OF SCATTERED KNOWLEDGE STRUCTURIZER

Considering that constructing various structured knowledge from a large scale of scattered information is a complex task that requires strong comprehension and generation abilities, and that LLMs have demonstrated a good capacity for integrating structured knowledge in previous work, we designed prompts to drive LLMs in achieving this process, as shown in Figure 7

### A.3    PROMPTS OF STRUCTURED KNOWLEDGE UTILIZER

Considering that questions in knowledge-intensive reasoning can be complex combinatorial tasks, breaking them down into multiple simpler sub-questions can leverage the structured knowledge more effectively for reasoning. Therefore, we designed prompts to drive LLMs to achieve question decomposition and precise knowledge extraction, as shown in Figure 6.

---

**Prompts in Constructing Table**

Instruction:
According to the requirements described in the Requirement section, extract the complete relevant tables from the Raw Content.
Note that when constructing the table, you should include the table title and source information, such as which company's report table comes from.
First, identify the keywords in the Requirement, including entity names and attribute names, and then use these keywords to extract information from the Raw Content. If the Raw Content does not contain the information required by the Requirement, extract only a small amount of information that is most relevant to the Requirement.
When analyzing the Requirement and extracting from the Raw Content, do not translate; maintain the original language.

Examples:
########
|Broadway Financial Corporation and Subsidiary|\n|---|\n|**Consolidated Statements of Financial Condition**|\n|**In thousands, except share and per share amounts**|\n|**Assets**|\n|Cash and due from banks|\$ 6,037|\n|Interest-bearing deposits in other banks|\$ 61,085|\n|Cash and cash equivalents|\$ 67,122|\n|Securities available-for-sale, at fair value|\$ 293,243|\n|Loans receivable held for investment, net of allowance of \$7,552 and \$7,348|\$ 926,497|\n|Accrued interest receivable|\$ 5,638|\n|Federal Home Loan Bank ("FHLB") stock|\$ 10,292|\n|Federal Reserve Bank ("FRB") stock|\$ 3,543|\n|Office properties and equipment, net|\$ 9,731|\n|Bank owned life insurance, net|\$ 3,286|\n|Deferred tax assets, net|\$ 9,827|\n|Core deposit intangible, net|\$ 2,027|\n|Goodwill|\$ 25,858|\n|Other assets|\$ 13,400|\n|Total assets|\$ 1,370,464|
########

Raw Documents:
{documents}

Query:
{query}

Output:

---

**Prompts in Constructing Graph**

Instruction:
According to the requirements described in the Requirement section, extract the necessary triples from the Raw Content.
The triples should be output on one line in the format: {{'head': '...', 'relation': '...', 'tail': ['...', '...']}}.
Note: Instead of extracting all triples from the text, analyze the relationships and entities mentioned in the Requirement and only extract the relevant triples.
Additionally, ensure that the 'head' and 'tail' in your output are as complete as possible. They can consist of more than just a single word or phrase—they may also be sentences or paragraphs of text. Aim to keep them consistent with the original text without any abbreviations.

Examples:
########
{{"head": "LLM4Vuln: A Unified Evaluation Framework for Decoupling and Enhancing LLMs\' Vulnerability Reasoning", "relation": "reference", "tail": ["Why Can GPT Learn In-Context? Language Models Implicitly Perform Gradient Descent as Meta-Optimizers.", "Can Large Language Models Be an Alternative to Human Evaluations?"]}}
########

Raw Documents:
{documents}

Query:
{query}

Output:

---

**Prompts in Constructing Algorithm**

Instruction:
Based on the requirements described in the Requirement section, extract the necessary algorithm pseudocode from the Raw Content.
You are required to follow the reasoning and output format provided in the Examples, ensuring that each action in the pseudocode is linked to specific information from the original document.
Note: If the requirements described cannot be resolved using pseudocode, do not force it. Instead, simply list the information that can address the requirements.

Examples:
########
Initialize components:
    Intel_CPU = "Intel high performance multi-core CPU"
    Huawei_CPU = "Huawei AI-powered multitasking CPU with thermal management"
    Apple_CPU = "Apple efficient CPU with seamless macOS integration"
    Huawei_Monitor = - Fan_120mm = "120mm standard cooling fan"
    Fan_140mm = "140mm high-efficiency fan"
Evaluate user requirements:
    if need high-performance CPU:
        IF user needs high refresh rate display:
            selected_CPU = Intel_CPU # According to original content, Intel CPU is high performance
########

Raw Documents:
{documents}

Query:
{query}

Output:

---

**Prompts in Constructing Catalogue**

Instruction:
According to the requirements described in the Requirement section, extract the necessary directory structure, which is a hierarchical summary. The levels and the number of nodes at each level should be determined based on the specific context.
You are required to follow the reasoning and output format provided in the Examples. Make sure that each level of the summary has a unique identifier for distinguishing different levels, and that each summary is detailed.
Note: You need to extract as much relevant information as possible from the Raw Content, including entity names and character names mentioned in the Requirement, to construct a complete directory structure.

Examples:
########
1.First-Level Summary 1: AI Technology and Regulatory Challenges
•The podcast explores the complex relationship between AI advancements and existing legal frameworks, with a particular focus on privacy laws like HIPAA and how they interact with technological innovation.
(1) Second-Level Summary 1: Regulatory Concerns in Financial Services
•Ethan Mollick highlights concerns that the current regulatory environment in financial services is not well-suited to address the unique challenges posed by AI, particularly the uncertainty surrounding the applicability of existing regulations.
•(a) Third-Level Summary 1: Innovation Hindered by Regulatory Ambiguity
•Mollick discusses how the lack of clarity in regulations impedes the ability of industries, like finance, to fully harness the potential of AI technologies.
########

Raw Documents:
{documents}

Query:
{query}

Output:

---

Figure 7: Prompts and examples used in data constructing for training the hybrid structure router, including synthesizing tasks, simulating solutions, and judging difference solutions.

---

**Prompts in Synthesizing**

---

Instruction:
In document question-answering tasks, there is a category of questions known as single-hop questions. The optimal strategy for solving this type of question is to split the document into multiple independent chunks and then select the most suitable chunk based on the query; for chain reasoning questions, such as finding citation and reference relationships among multiple given papers, the optimal strategy is to present the information in the document in the form of a graph; for statistical questions, such as analyzing and comparing financial data of multiple companies, the optimal strategy is to present the information in the document in the form of a table; for configuration questions, such as needing to assemble a computer from a large batch of different models and components based on specific user needs, the optimal strategy is to present the information in the document in the form of an algorithm; for summary questions, such as summarizing a large-scale meeting record into a meeting summary, the optimal strategy is to organize the meeting records into a catalogue format.

Requirements:
A diversity of content needs to be generated, covering various fields and scenarios.
1.Adhere to the approach indicated in the Examples, that is, when performing question-answering tasks on such document collections, the optimal strategy is to split the documents into multiple independent chunks.
2.Use DOCUMENTS_INFO and QUERY as markers, with ## as a separator between each output.

Examples:
########
DOCUMENTS_INFO:
"Beethoven's Life", "Einstein's Life", "Newton's Life", "Da Vinci's Life", "Galileo's Life", "Voltaire's Life"
QUERY:
In which year was Newton born according to the given document collection?
##
DOCUMENTS_INFO:
2020 Financial Report of Netflix 2020 Financial Report of Alibaba 2020 Financial Report of Tencent
QUERY:
classify the companies in the above documents according to revenue, with more than 100 billion as high-income companies and less than 100 billion as low-income companies.
##
DOCUMENTS_INFO:
RAG: Retrieval-Augmented Generation for Knowledge-Intensive NLP Tasks DPR: Dense Passage Retrieval for Open-Domain Question Answering BERT: Pre-training of Deep Bidirectional Transformers for Language Understanding T5: Exploring the Limits of Transfer Learning with a Unified Text-to-Text Transformer RoBERTa: A Robustly Optimized BERT Pretraining Approach ALBERT: A Lite BERT for Self-supervised Learning of Language Representations
QUERY:
decide the reference and citation relationship among the given documents
########

Output:

---

**Prompts in Simulating**

---

Instruction:
You need to provide multiple possible solutions based on the given problem, with each solution utilizing a different type of knowledge.

Requirements:
1.The types of knowledge include three: chunk, graph, table, algorithm, catalogue
2.Your proposed solutions need to clarify how to construct the document from the current task into the corresponding type of knowledge, and how to use that type of knowledge to answer the questions in the task.
3.Follow the output format in the Examples, with each solution on a separate line, and each solution beginning with the chosen type of knowledge.

Examples:
########
Query:
Based on the documents above, summarize the development trends of this company over the past two years.
Output:
1.Chunk: To build and use chunk-type knowledge, I first need to break these financial reports into multiple independent chunks. Then, based on the query, I will select a few chunks that best reflect the company's development trends. Finally, I will summarize the company's development trends over the past two years based on the information from the selected chunks.
2.Graph: To build and use graph-type knowledge, I need to present the information from the documents in the form of a graph, with the company name as the head node, financial metrics as the relationships, and corresponding financial data as the tail nodes. Then, I will find the relevant company and subgraph based on the query and summarize the company's development trends over the past two years based on the information in the subgraph.
3.Table: To build and use table-type knowledge, I need to present the information from the documents in table form, with the company name as the rows and various financial metrics as the columns. The value in each cell will be the corresponding financial data. Finally, I will summarize the company's development trends over the past two years based on the information in the table.
4.Algorithm: To build and use algorithm-type knowledge, I need to use the company's metrics as judgment criteria, with each decision in the process determining whether to consider the information from that metric. Ultimately, I will summarize the company's development trends based on this process.
5.Catalogue: To build and use catalogue-type knowledge, I need to classify the company's different financial metrics by importance. Each level of the directory will include specific information about the metric, and I will draw conclusions based on all the metric information in this catalogue.
########

Query:
{query}

Output:

---

**Prompts in Judging**

---

Instruction:
For the given task, there are currently multiple solutions available, each utilizing a different type of knowledge. You need to rank the effectiveness of these solutions and identify the most suitable one. First present a comparison and analysis of each solution, and then output the final ranking results, using '>', '<', and '=' to indicate the ranking relationships.

Documents Info:
{documents_info}

Query:
{query}

Solutions:
{solutions}

Output:

Figure 8: Prompts used in convert raw information in original documents into structured knowledge via LLMs, different type of structured knowledge use different prompts, and the chunk is directly by splitting without prompt.

