# OpenReview forum: "StructRAG: Boosting Knowledge Intensive Reasoning of LLMs via Inference-time Hybrid Information Structurization"
_ICLR.cc/2025/Conference — ICLR 2025 Poster_

### Official Review · Reviewer_QD9m · 2024-10-31

**Soundness:** 4
**Presentation:** 4
**Contribution:** 3
**Rating:** 8
**Confidence:** 4

**Summary:**

The paper addresses the challenge of handling RAG with complex documents, where information may be scattered across different parts of multiple documents. For RAG to be accurate, it must precisely retrieve and integrate all relevant texts, which presents a significant challenge. To tackle this, the authors propose a method that mimics human cognitive processes by using LLMs to convert dispersed information into various types of structured knowledge. Additionally, they employ a Hybrid Structure Router, trained with DPO, to select the appropriate structure type for each query. The implementation of this approach demonstrates its effectiveness in improving performance across most tasks.

**Strengths:**

**Originality**: This paper introduces a novel approach to addressing the complex issue of RAG from multiple knowledge sources. The proposed process includes summarizing structured knowledge into five categories and utilizing LLMs for the construction and retrieval of this structured knowledge. This approach demonstrates a high degree of originality.

**Quality**: The writing quality of the paper is commendable, with coherent reasoning and a logical flow that makes it easy for readers to understand. The authors have conducted detailed experiments across multiple datasets, effectively demonstrating the validity of StructRAG.

**Clarity**: The paper is logically and structurally well-organized, with a clear presentation of the proposed framework's structure and processes. The use of formulas and diagrams effectively illustrates the concepts and workflow.

**Significance**: This paper effectively addresses the challenge of handling RAG with complex documents by aggregating knowledge from different sources and structuring it into specified formats. This approach significantly enhances the performance and applicability of RAG systems.

**Weaknesses:**

The paper, while introducing the StructRAG framework, relies significantly on the inherent capabilities of large language models (LLMs) for structuring and utilizing raw knowledge. However, it could benefit from a more detailed discussion on the challenges encountered during knowledge processing and on strategies for more effectively leveraging LLMs in this context.

**Questions:**

1. Could you provide more detailed examples of tasks that are suitable for each of the five types of structured knowledge you propose? This would help clarify the rationale behind categorizing them into these five types.

2. When using LLMs to extract structured knowledge, do the LLMs have the capability to directly generate knowledge in the specified formats? Did you perform any checks or corrections on the outputs generated by the LLMs?

---

> ### Author Response · Authors · 2024-11-25
> **Response to Reviewer QD9m**
>
> Thank you for your thorough review and insightful feedback on our paper. Your comments reflect a deep understanding of the field, and we are happy to engage in a discussion on the issues you raised.
>
> > **Could you provide more detailed examples of tasks that are suitable for each of the five types of structured knowledge you propose? This would help clarify the rationale behind categorizing them into these five types.**
>
> Thanks for your helpful suggestion. We provide detailed examples for each of the five types of structured knowledge as follows:
>
> - Statistical questions often use **table** structure. For example, a question might be: "*Please categorize the aforementioned companies based on 'Cash Flow from Investing Activities' into the following groups: High Profit (above 100,000,000.00), Medium Profit (greater than 0 and up to 100,000,000.00), and Low Profit (0 and below).*" The documents might include financial data from dozens companies.
> - Chain reasoning problems typically use **graph** structure. For instance, a question might be: "*We kindly ask you to thoroughly review the provided papers and construct a citation chain from them.*" The documents might consist of the full texts of 20 papers.
> - Text matching problems often use **chunk** structure. For example, a question might be: "*Review the above judicial documents and determine which case type each judgment is associated with: ['Execution Case - Civil', 'Inheritance Dispute', 'Intellectual Property and Competition'].*" The documents might include dozens of judicial judgments.
> - Summarization problems typically use **catalogue** structure. For instance, a question might be: "*What is the core topic of this meeting?*" The documents might consist of a complete meeting transcript.
> - Action planning problems often use **algorithm** structure. For example, a question might be: "*How do I assemble a desktop computer with high image quality and long battery life, with no special requirements for speed?*" The documents might include manuals for various components such as monitors, graphics cards, CPUs, and power supplies.
> We will provide more details of these examples in the revision of our paper to offer further clarity.
>
> > **When using LLMs to extract structured knowledge, do the LLMs have the capability to directly generate knowledge in the specified formats? Did you perform any checks or corrections on the outputs generated by the LLMs?**
>
> This is an insightful question. By observing the model’s outputs, we found that, in most cases, the LLM is able to produce the specified format. In this work, we did not make corrections to the model’s outputs. However, as mentioned in the case study section of the paper (Section 5.4.3), the structured knowledge generated by the LLM may deviate from the original document (Figure 5). We will include a more detailed discussion on this point in the revision. Additionally, to address this issue, we plan to explore training the model with algorithms such as DPO in future work, with the goal of further enhancing the reliability of our StructRAG framework.

---

### Official Review · Reviewer_EKh5 · 2024-11-03

**Soundness:** 3
**Presentation:** 3
**Contribution:** 3
**Rating:** 5
**Confidence:** 3

**Summary:**

This paper proposed a novel method to summarise the document with Hybrid Structure Router, and it dynamically converted documents into graph, chunk, catalogue, algorithm, table. It showed that the new method had better performance in terms of score and latency. The paper contains thorough literature review, and detailed experiment results. However, the paper only evaluates the performance on Qwen2-7B-Instruct model, which is a relatively small model. It did not evaluate on other main stream models like llama, and also did not evaluate the new method on models with larger amount of parameters. For example, the GraphRAG method used GPT-4-turbo in their original paper, and it is very hard to conclude that StructRAG is better than GraphRAG with the results of Qwen2-7B-Instruct presented in this paper.

**Strengths:**

- It proposed a novel method
- It conducted thorough literature review
- It compared the method with other RAG methods including Long-context, RAG, RQ-RAG, GraphRAG

**Weaknesses:**

- It did not provide the definition of LLM score
- It only tested the new model with Qwen2-7B-Instruct with relatively small amount of parameters, and lacks of experiment results with other models or models with larger amount of parameters

**Questions:**

- What is the definition of LLM score
- Did you tested the new method StructRAG with other models, or models with larger amount of parameters?

---

> ### Author Response · Authors · 2024-11-25
> **Response to Reviewer EKh5**
>
> Thank you for your thoughtful review and feedback. We appreciate the time and effort you invested in evaluating our work. Sorry for the late reply, we spent a significant amount of time supplementing the experiments on LLaMA-3.1-70B-Instruct. Below, we address the concerns you raised:
>
> > **What is the definition of LLM score?**
>
> The LLM score is the official metric used in the Loong benchmark [1], which leverages GPT-4 to assess the correctness of model-generated content based on the golden answer and the question’s requirements, including accuracy, hallucinations, and completeness, with a percentage score ranging from 0 to 100. In our experiments, we strictly follow the scoring code provided in the Loong benchmark to ensure the reliability of the experimental results. Additional details about the LLM score will be included in the revision.
>
> [1] Leave No Document Behind: Benchmarking Long-Context LLMs with Extended Multi-Doc QA. In EMNLP 2024. https://aclanthology.org/2024.emnlp-main.322.pdf
>
> > **It only tested the new model with Qwen2-7B-Instruct with relatively small amount of parameters, and lacks of experiment results with other models or models with larger amount of parameters**
>
> There might be a slight misunderstanding here. The StructRAG framework consists of three components: Router, Structurizer, and Utilizer. The Router is trained based on the Qwen2-7B-Instruct model, while both the Structurizer and Utilizer directly use Qwen2-72B-Instruct as their base model.
>
> > **For example, the GraphRAG method used GPT-4-turbo in their original paper, and it is very hard to conclude that StructRAG is better than GraphRAG with the results of Qwen2-7B-Instruct presented in this paper.**
>
> Due to the high cost of using GPT-4-turbo, we employ Qwen2-72B-Instruct as the foundation for the Structurizer and Utilizer modules in our experiments. To ensure fairness, we also use Qwen2-72B-Instruct as the foundation model for all baselines. As a result,  our experiments demonstrate that StructRAG outperforms GraphRAG when both are built on the same base model.
>
> > **Did you tested the new method StructRAG with other models, or models with larger amount of parameters?**
>
> To further address your concerns, we replace the base model of StructRAG and all baselines from Qwen2-72B-Instruct to LLaMA-3.1-70B-Instruct. The experimental results are shown in the table below, demonstrating that StructRAG continues to achieve SOTA performance overall. This further confirms that StructRAG is a general framework capable of outperforming baselines across different base models.
>
> Table. Overall score on Loong benchmark for StructRAG and all baselines based on LLaMA-3.1-70B-Instruct
>
> | Method          |    Set 1   |           |    Set 2   |           |    Set 3   |           |    Set 4   |           |
> |-----------------|----------|---------|----------|---------|----------|---------|----------|---------|
> |                 |  LLM Score |     EM    |  LLM Score |     EM    |  LLM Score |     EM    |  LLM Score |     EM    |
> | Long-context    |   42.74    |   0.17    |   36.68    |   0.08    |   33.66    |   0.06    |   30.87    |   0.04    |
> | RAG             |   31.27    |   0.04    |   31.96    |   0.04    |   26.79    |   0.05    |   23.91    | **0.06** |
> | RQ-RAG          |   35.64    |   0.12    |   32.21    |   0.07    |   23.36    |   0.06    |   23.04    |   0.01    |
> | GraphRAG        |   35.45    |   0.09    |   31.76    |   0.06    |   26.00    |   0.04    |   18.95    |   0.00    |
> | StructRAG(Ours) | **60.22** | **0.25** | **43.81** | **0.11** | **38.98** | **0.10** | **35.61** |   0.05    |

---

> ### Author Response · Authors · 2024-11-27
> **Looking forward to your feedback**
>
> Dear Reviewer EKh5,
>
> Thanks for your time and valuable feedback on our submission. We are writing to kindly remind you that we have responded to your questions and would appreciate it if you could review our response when you have the opportunity. If you have any further questions or concerns regarding our explanation, please feel free to reach out to us. Your feedback is invaluable for refining our work, and we look forward to hearing from you.
>
> Thank you again for your time !
>
> Sincerely
> All authors

---

> > ### Comment · Reviewer_EKh5 · 2024-12-01
> >
> > Thanks for adding the additional data with Llama-3.1-70B-Instruct.

---

### Official Review · Reviewer_WHkD · 2024-11-03

**Soundness:** 3
**Presentation:** 3
**Contribution:** 3
**Rating:** 8
**Confidence:** 4

**Summary:**

This paper introduces StructRAG, a novel framework for improving retrieval-augmented generation (RAG) in knowledge-intensive reasoning tasks. The key insight is that existing RAG methods struggle when useful information is scattered across documents, making it difficult to identify and reason with key information.

Inspired by cognitive theories about how humans process information using different knowledge structures, StructRAG consists of three main components:

1. A hybrid structure router that determines the most appropriate structure type (e.g., tables, graphs) for a given task
2. A scattered knowledge structurizer that converts raw documents into the chosen structured format
3. A structured knowledge utilizer that decomposes complex questions and performs reasoning using the structured knowledge

To train the hybrid structure router effectively, the authors develop a novel pipeline for generating training data through task synthesis, solution simulation, and preference judgment, combined with the DPO algorithm for preference learning.

Experimental results show that StructRAG achieves state-of-the-art performance across various knowledge-intensive tasks, with particularly strong improvements on more complex tasks. The framework also demonstrates better performance and faster operation compared to existing Graph RAG methods.

The paper's main contribution is presenting a cognitively-inspired approach to structuring information during inference time in RAG systems, offering a promising direction for handling complex reasoning tasks where relevant information is scattered across multiple sources.

**Strengths:**

This paper demonstrates several notable strengths:

- The novel integration of cognitive science principles into RAG systems. The authors establish meaningful connections between human information processing and computational approaches, making the cognitive science foundation both natural and practical.

- The paper addresses a well-recognized challenge in the field: efficient information retrieval from unstructured inputs. Their solution, particularly the Hybrid Structure Router that mimics human decision-making combined with DPO fine-tuning, offers an innovative yet implementable approach.

- The experimental evaluation is comprehensive, comparing StructRAG against current state-of-the-art approaches including long-context models, traditional RAG, RQ-RAG, and GraphRAG. The promising results across these various settings demonstrate the method's practical value.

- The ablation studies are particularly insightful, reflecting the authors' thorough analysis and iterative refinement of their methodology.

**Weaknesses:**

While not a significant weakness, one suggestion concerns the comparison with Graph RAG. Graph RAG typically excels with complex relational data rather than long-context scenarios, which might explain its relatively poor performance in many of the experimental settings. A more targeted comparison using relationship-heavy datasets could provide additional insights into the relative strengths of both approaches.

**Questions:**

Lines 458-460 highlight a challenge in real-world applications. How do the authors propose to address this challenge?

---

> ### Author Response · Authors · 2024-11-25
> **Response to Reviewer WHkD**
>
> Thank you for your thorough review and insightful feedback on our paper. Your comments reflect a deep understanding of the field, and we are happy to engage in a discussion on the issues you raised.
>
> > **Graph RAG typically excels with complex relational data rather than long-context scenarios, which might explain its relatively poor performance in many of the experimental settings. A more targeted comparison using relationship-heavy datasets could provide additional insights into the relative strengths of both approaches.**
>
> Thank you for your constructive suggestion. Comparing with GraphRAG on relationship-heavy datasets is indeed valuable. However, applying both StructRAG and GraphRAG to datasets with extensive inter-document relationships requires substantial computational resources. This is because StructRAG relies on LLM to extract structured knowledge from input documents, while GraphRAG relies on LLM to generate triples from the input documents and reorganize them. When dealing with large-scale document collections, these processes can result in significant computational overhead. Nonetheless, we will consider attempting this experiment in future work.
>
> > **Lines 458-460 highlight a challenge in real-world applications. How do the authors propose to address this challenge?**
>
> In this paper, we rely solely on the capabilities of the LLM itself to implement the Structurizer. As a result, the process of constructing structured knowledge may introduce discrepancies with the original document due to the limitations of the LLM's capabilities. To address this issue, a potential future direction is to train LLM using algorithms like DPO, thereby enhancing its ability to construct structured knowledge.

---

> > ### Comment · Reviewer_WHkD · 2024-11-27
> >
> > Thanks for your responses. I believe the deeper research on pros and cons compared with GraphRAG could be valuable.

---

### Official Review · Reviewer_Q2p8 · 2024-11-04

**Soundness:** 3
**Presentation:** 3
**Contribution:** 3
**Rating:** 6
**Confidence:** 4

**Summary:**

The paper introduces StructRAG, a novel framework designed to enhance LLMs for knowledge-intensive reasoning tasks by structuring information at inference time. StructRAG addresses limitations in existing retrieval-augmented generation (RAG) methods, which struggle when relevant knowledge is dispersed and reasoning demands are complex. Inspired by cognitive theories, StructRAG applies a hybrid structure router to identify the best structure type for each task, a scattered knowledge structurizer to transform raw documents into structured knowledge, and a structured knowledge utilizer to perform multi-step reasoning. This framework achieves state-of-the-art results across multiple benchmarks and tasks, particularly excelling in challenging scenarios where information is fragmented and extensive reasoning is required.

**Strengths:**

The paper addresses a challenging problem in LLM and RAG research, focusing on improving knowledge-intensive reasoning for real-world tasks. The idea of information structurization for RAG is clear and innovative, offering a new approach in this area.

The architecture and its components are described clearly, with each part of the StructRAG model explained in a way that makes it easy to follow. The experiments are thorough and well-documented, including ablation studies that show the contributions of each module and support the effectiveness of the approach.

Overall, the work has practical applications and could be adapted to a range of RAG-related tasks, suggesting it has potential for further impact in the field.

**Weaknesses:**

The study presents StructRAG’s effectiveness with a single structured knowledge type for each task, yet it is unclear how the model would perform on tasks requiring multiple types simultaneously. For example, an ablation study that evaluates combinations of two fixed structure types could shed light on whether using multiple structure types consistently improves performance across different task categories.

The paper does not provide details on the distribution of structure types chosen by the hybrid structure router in its dataset. Since some structure types, such as algorithmic pseudo code, may be more complex to extract compared to tables or catalogues, reporting this distribution would clarify the router’s efficacy across different structure types. Additionally, an analysis of the structurizer’s performance in cases involving algorithmic structuring would help assess whether these cases yield similar performance gains as simpler structures.

Table 5 highlights performance differences when using fixed structure types, but these differences may be influenced by the distribution of tasks that align better with certain types (e.g., most tasks in the dataset naturally suited to tables). A breakdown of the dataset showing the router’s selected structure types and a comparison to human-selected types would be useful. Such a comparison could indicate how well the router aligns with human judgment and whether adding certain structure types as a standard complement might enhance overall performance in diverse datasets.

**Questions:**

In Table 3 and Figure 3, the exact match (EM) rate for Set 4 appears close across methods or even equivalent in some cases. Could the authors clarify whether this is due to specific characteristics of the dataset (e.g., limited answer variation) or if this is a coincidence? If the dataset impacts EM rates, additional context would be helpful.

---

> ### Author Response · Authors · 2024-11-25
> **Response to Reviewer Q2p8**
>
> Thank you for reviewing our manuscript and providing constructive feedback. We have reviewed your comments carefully and prepared responses to address your concerns.
>
> > **The study presents StructRAG’s effectiveness with a single structured knowledge type for each task, yet it is unclear how the model would perform on tasks requiring multiple types simultaneously.**
>
> Thank you for your constructive suggestion. You raise an important point about how StructRAG would perform on tasks requiring multiple types of structured knowledge simultaneously. While our current study focuses on a single knowledge type per task, exploring multi-type tasks is indeed valuable. This would offer deeper insights into the model’s flexibility in handling more complex scenarios. We agree that this is a worthwhile avenue for future work and plan to explore it further.
>
> > **The paper does not provide details on the distribution of structure types chosen by the hybrid structure router in its dataset.**
>
> Thank you for pointing this out. We conduct a statistical analysis of the structured knowledge types selected by the hybrid structure router. We find that, in the Loong dataset, the router predominantly selects three main types, including chunk, table and graph. This is primarily due to the limited variety of task types within the benchmark. Therefore, a benchmark with a broader range of diverse tasks would be valuable. We will include the relevant details and analysis in the revised version of the paper.
>
> > **A comparison to human-selected types would be useful. Such a comparison could indicate how well the router aligns with human judgment.**
>
> Thank you for your valuable suggestion. We agree that a comparison to human-selected types would be useful, as it could provide insights into how well the router aligns with human judgment. However, due to time constraints, we are unable to perform human annotation in this study. Nevertheless, we plan to address this in future work by conducting a more thorough analysis and comparison.
>
> > **In Table 3 and Figure 3, the exact match (EM) rate for Set 4 appears close across methods or even equivalent in some cases.**
>
> In the Loong benchmark, achieving an exact match between the model's output and the golden answer is highly challenging [1], particularly under the most complex settings with much longer documents, such as Set 4. Therefore, as shown in Table 3 and Figure 3, the EM scores of different methods are almost equally low, as they can only achieve exact matches in relatively simple cases. We will include further discussion on this point in the revision and explore ways to improve the EM score in future work.
>
>
> [1] Leave No Document Behind: Benchmarking Long-Context LLMs with Extended Multi-Doc QA. In EMNLP 2024. https://aclanthology.org/2024.emnlp-main.322.pdf

---

> > ### Comment · Reviewer_Q2p8 · 2024-11-27
> >
> > Thank you very much for your detailed explanations. I believe the future work mentioned in the response would be important to make it a comprehensive study.

---

### Meta-Review · Area_Chair_MEAv · 2024-12-15

**Metareview:**

This paper proposes a novel framework for solving knowledge-intensive reasoning tasks, which are difficult for existing retrieval-augmentation generation methods to address in locating key information. The framework comprises three components: a structure router for structure type selection, a knowledge structurizer for knowledge extraction, and a knowledge utilizer for answer generation. This framework achieves state-of-the-art performance across various tasks, and each component's effectiveness is demonstrated through ablation studies.

The standout design in this framework is the structure router, which contributes the most to performance gains. The method of constructing synthetic data for training the router is innovative, effectively addressing the issue of data scarcity. Furthermore, the significant performance improvements observed in the experimental results demonstrate the effectiveness of StructRAG in knowledge-intensive tasks.

One area for potential improvement is that the effectiveness of StructRAG depends on the underlying language models used for structurization. An additional study on structurization across various language models would be valuable for broader application.

**Additional Comments On Reviewer Discussion:**

The author’s responses are reasonable. Some suggested analyses by the reviewers could be included in the revised version.

---

### Decision · Program_Chairs · 2025-01-22

Accept (Poster)